# FINE: Future-Aware Inference for Streaming Speech Translation

## Abstract

A popular approach to streaming speech translation is to employ a single offline model together with a *wait-k* policy to support different latency requirements. It is a simpler alternative compared to training multiple online models with different latency constraints. However, there is an apparent mismatch in using a model trained with complete utterances on partial streaming speech during online inference. We demonstrate that there is a significant difference between the speech representations extracted at the end of a streaming input and their counterparts at the same positions when the complete utterance is available. Built upon our observation that this problem can be alleviated by introducing a few frames of future speech signals, we propose **F**uture-aware **in**ferenc**e** (FINE) for streaming speech translation with two different methods to make the model aware of the future. The first method FINE-Mask incorporates future context through a trainable masked speech model. The second method FINE-Wait simply waits for more actual future audio frames at the cost of extra latency. Experiments on the MuST-C EnDe, EnEs, and EnFr benchmarks show that both methods are effective and can achieve better trade-offs between translation quality and latency than strong baselines, and a hybrid approach combining the two can achieve further improvement. Extensive analyses suggest that our methods can effectively alleviate the aforementioned mismatch problem between offline training and online inference.

## 1 Introduction

Streaming speech translation (ST) systems consume audio frames incrementally and generate real-time translations, unlike their offline counterparts which have access to the complete utterance before starting to translate. Because of the streaming nature, streaming ST models commonly use unidirectional encoders (Ren et al., 2020; Ma et al., 2020b; Zeng et al., 2021) and are trained with some *wait-k* policy (Ma et al., 2019) that determines whether to wait for more speech frames or emit target tokens. In real-world applications, however, it is a costly effort to train and maintain multiple models to satisfy different latency requirements (Zhang & Feng, 2021). Recently, some works (Papi et al., 2022; Dong et al., 2022) show that offline models can be adapted to streaming scenarios with *wait-k* policies to meet different latency requirements and achieve comparable or better performance, partially due to their use of more powerful bidirectional encoders. However, there is an inherent mismatch in using a model trained with complete utterances on incomplete streaming speech during online inference (Ma et al., 2019).

Intuitively, speech representations extracted from streaming inputs (Figure 1(b)) are less informative than in the case with full speech encoding (Figure 1(a)). Two questions arise naturally: how much is the difference in speech representations between the two inference modes, and is it significant enough to cause problems? We analyze the gap in speech representations, measured by cosine similarity, at different positions in the streaming input compared to using the full speech (Section 3). We find that there is a significantly greater gap for representations closer to the end of a streaming segment, with an average similarity score as low as 0.2 for the last frame, and the gap quickly narrows for frames further away. Moreover, we observe more degradation in translation quality for utterances with the greatest gap in speech representations between online and offline inference.

Based on the above findings, we hypothesize that the lack of future contexts at the end of streaming inputs could be detrimental to streaming speech translation. To this end, we propose two novel

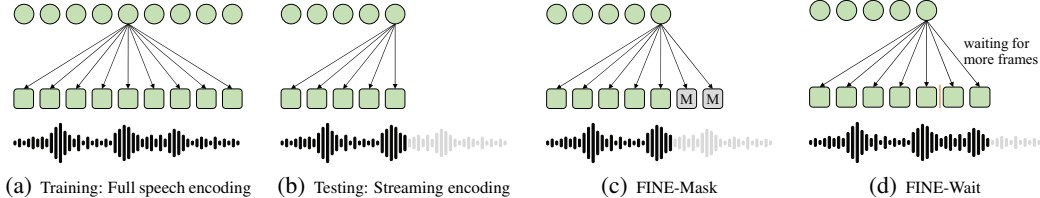

(a) Training: Full speech encoding    (b) Testing: Streaming encoding    (c) FINE-Mask    (d) FINE-Wait

Figure 1: (a) and (b) represent the input mismatch between offline training and streaming testing. (c) and (d) denote the proposed methods FINE-Mask and FINE-Wait, respectively. "M" in (c) denotes the mask token. Our methods introduce more informative future context to mitigate the mismatch.

**F**uture-aware **in**ferenc**e** (FINE) strategies for streaming speech translation: FINE-Mask and FINE-Wait, as shown in Figure 1(c) and 1(d). In FINE-Mask, we append a few mask embeddings to the end of the current streaming speech tokens as additional input to the acoustic feature extractor, which based on its masked modeling capability can implicitly estimate and construct future contexts in the corresponding hidden representations and extract more accurate representations for the frames in the streaming input. Since we find that only the speech representations of the last few positions in the streaming input are severely affected by the mismatch problem, the closest future context could provide the most improvement. Thus in FINE-Wait, we simply wait for a few extra speech tokens during streaming encoding and use them as the future context to extract improved representations for the frames in the original streaming segment. FINE-Wait incurs additional latency as the strategy requires waiting for more oracle future context, but it achieves significant improvement in translation quality and leads to a better trade-off.

We conduct experiments on the MuST-C EnDe, EnEs, and EnFr benchmarks. Experimental results show that our methods outperform several strong baselines on the trade-off between translation quality and latency. In particular, in the lower latency range (when AL is less than 1000*ms*), we achieve improvements of 8 BLEU in EnDe, 12 BLEU in EnEs, and 6 BLEU in EnFr. Extensive analyses demonstrate that introducing future context reduces the representation gap between the full speech encoding and the partial streaming encoding.

## 2 BACKGROUND

Speech translation systems can be roughly categorized into non-streaming (offline) and streaming (online) depending on the inference mode. Regardless of the inference mode, speech translation models typically employ the encoder-decoder architecture and are trained on an ST corpus $\mathcal{D} = \{(\mathbf{x}, \mathbf{z}, \mathbf{y})\}$, where $\mathbf{x} = (x_1, \ldots, x_T)$ denotes an audio sequence, $\mathbf{z} = (z_1, \ldots, z_I)$ and $\mathbf{y} = (y_1, \ldots, y_J)$ the corresponding source transcription and target translation respectively.

**Non-Streaming Speech Translation** For the non-streaming ST task, the encoder maps the entire input audio $\mathbf{x}$ to the speech representations $\mathbf{h}$, and the decoder generates the $j$-th target token $y_j$ conditional on the full representations $\mathbf{h}$ and the previously generated tokens $y_{<j}$. The decoding process of non-streaming ST is defined as:

$$p(\mathbf{y} \mid \mathbf{x}) = \prod_{j=1}^{J} p\left(y_j \mid \mathbf{x}, \mathbf{y}_{<j}\right). \qquad (1)$$

A significant amount of work has focused on non-streaming ST, including pre-training (Wang et al., 2020a; Dong et al., 2021a; Tang et al., 2022; Ao et al., 2022),multi-task learning (Liu et al., 2020; Indurthi et al., 2020; 2021),data augmentation (Pino et al., 2019; Di Gangi et al., 2019b; McCarthy et al., 2020),knowledge distillation (Dong et al., 2021b; Zhao et al., 2021; Du et al., 2022),and cross-modality representation learning (Tang et al., 2021; Fang et al., 2022; Ye et al., 2022).

**Streaming Speech Translation** A streaming ST model generates the $j$-th target token $y_j$ based on streaming audio prefix $\mathbf{x}_{\leq g(j)}$ and the previous tokens $y_{<j}$ , where $g(j)$ is a monotonic non-

decreasing function representing the ending timestamp of the audio prefix that needs to be consumed to generate the $j$-th word. The decoding probability is calculated as:

$$p(\mathbf{y} \mid \mathbf{x}) = \prod_{j=1}^{J} p\left(y_j \mid \mathbf{x}_{\leq g(j)}, \mathbf{y}_{<j}\right). \tag{2}$$

Thus, a streaming ST model requires a policy to determine whether to wait for more source speech or emit new target tokens. Recent studies (Ma et al., 2020b; Ren et al., 2020; Zeng et al., 2021; Dong et al., 2022) on streaming ST make read/write decisions based on a variant of the *wait-k* policy (Ma et al., 2019) that was initially proposed for streaming text translation, which alternates write and read operations after reading the first $k$ source tokens. Because there is no explicit word boundaries in a streaming audio, several works attempt to detect word boundaries in the audio sequence using methods such as fixed length (Ma et al., 2020b), Connectionist Temporal Classification (Ren et al., 2020; Zeng et al., 2021; Papi et al., 2022), ASR outputs (Chen et al., 2021), and integrate-and-fire (Dong et al., 2022). The *wait-k* policy is applied based on detected words rather than audio frames. In other words, $g(j)$ in Eq.(2) represents the length of audio segment corresponding to the first $j + k - 1$ detected words in the streaming ST. Moreover, some studies (Arivazhagan et al., 2019; Ma et al., 2020c; Zhang et al., 2020; Schneider & Waibel, 2020; Miao et al., 2021; Zhang & Feng, 2022a;b; Zhang et al., 2022; Chang & Lee, 2022; Liu et al., 2021) explore adaptive policies to dynamically decide when to read or write for streaming text and/or streaming speech translation. Several works (Zhang et al., 2021; Zhang & Feng, 2022c) apply knowledge distillation and fill future source positions with positional encoding to introduce future information during training for simultaneous machine translation within the prefix-to-prefix framework. In this paper, we focus on a matter less attended to – how to alleviate the mismatch between offline training and online inference.

## 3 PRELIMINARY ANALYSIS

In this section, we analyze the major mismatch in Transformer-based (Vaswani et al., 2017) ST architecture between offline training and online decoding. In full-sentence ST, the speech representation of each frame is obtained by attending to all unmasked frames by the multi-head attention in the transformer encoder layers. However, if directly applied to the streaming inference with the model trained offline, the speech representation of the current last frame will deteriorate because it can only attend to its previous frames. Recently, a common approach in speech translation is to stack a pre-trained Wav2Vec2.0 (Baevski et al., 2020) as the acoustic encoder with a semantic NMT encoder-decoder, and achieves SOTA performance in the ST task (Han et al., 2021; Dong et al., 2022; Fang et al., 2022; Ye et al., 2022), because it has been shown that a better speech representation can be learned via Wav2Vev2.0 (Baevski et al., 2020).

To explore the precise effects of streaming inputs, we first follow MoSST (Dong et al., 2022) to train an offline ST model on the MuST-C EnDe training set, where the acoustic encoder Wav2Vec2.0 is trainable. After the offline ST training, we conduct an analysis on the MuST-C EnDe tst-COMMON set. We remove the outliers and the noisy data, and select audios with a duration between 2s and 10s, resulting in a total of 1829 examples.

For an input sequence of audio frames $\mathbf{x} = (x_1, \ldots, x_T)$, the convolutional subsampler of Wav2Vec2.0 shrinks the length of the raw audio by a factor 320 and outputs the full speech representation sequence $\mathbf{a}$. In other words, every 320 elements in $\mathbf{x}$ become a vector in $\mathbf{a}$. For readability reasons, we uniformly use the notation $T$ to denote the sequence length of $\mathbf{a}$, i.e., $\mathbf{a} = (a_1, \ldots, a_T)$. This simplified notation does not undermine any of our conclusions while at the same time making the equations for readable[1]. For streaming input $\forall t \leq T, \hat{\mathbf{x}}_t = (x_1, \ldots, x_t)$, Wav2Vec2.0 will output the representation $\hat{\mathbf{a}}_t = (\hat{a}_{t,1}, \ldots, \hat{a}_{t,t})$.

### 3.1 WHICH PART OF STREAMING SPEECH REPRESENTATION IS WORSE?

To measure the gap of the speech representations between the offline and online inputs, we calculate the cosine similarity $s_{t,t'}$ between the speech representation at the $t'$-th ($t' \leq t$) position in the $t$-th

---

[1]Because we can always define $\mathbf{x} = (x_{1:T})$ such that $x_t$ represents consecutive 320 audio frames.

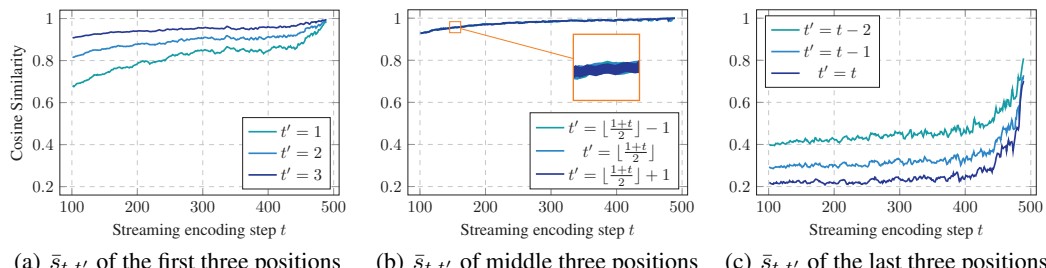

(a) $\bar{s}_{t,t'}$ of the first three positions  (b) $\bar{s}_{t,t'}$ of middle three positions  (c) $\bar{s}_{t,t'}$ of the last three positions

Figure 2: The average cosine similarity $\bar{s}_{t,t'}$ of the first three ($t' = 1, 2, 3$), middle three ($t' = \lfloor \frac{1+t}{2} \rfloor - 1, \lfloor \frac{1+t}{2} \rfloor, \lfloor \frac{1+t}{2} \rfloor + 1$), and last three ($t' = t - 2, t - 1, t$) positions for each encoding step $t$.

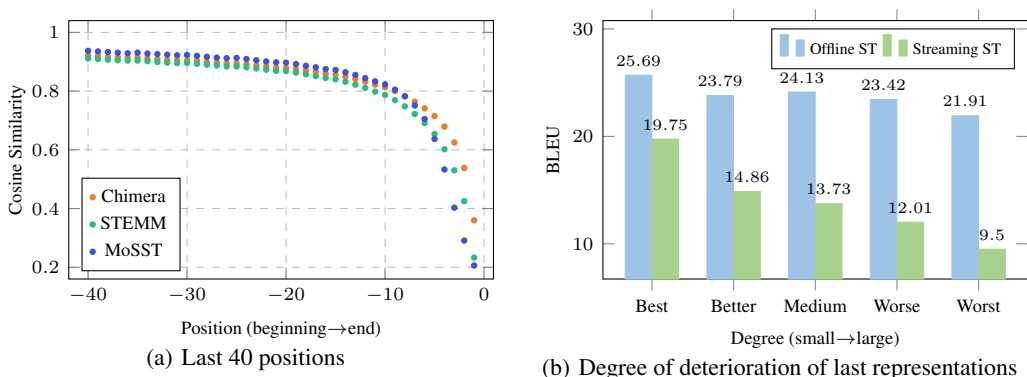

(a) Last 40 positions  (b) Degree of deterioration of last representations

Figure 3: (a). The average cosine similarity $\bar{s}_{t'}$ of the forty representations at the end positions (position$= -1$ denotes last position) in the streaming speech. (b). Performance with degree of deterioration of the representation at the last position of the streaming speech.

streaming audio input $\hat{\mathbf{x}}_t$ and the speech representation at the same position in the full encoding. Then we average the cosine similarities over the sentences in dataset $\mathcal{B}$ to obtain robust statistics.

$$\text{For } t' \leq t, \quad \bar{s}_{t,t'} = \frac{1}{|\mathcal{B}_t|} \sum_{\mathbf{x} \in \mathcal{B}_t} s_{t,t'}(\mathbf{x}) = \frac{1}{|\mathcal{B}_t|} \sum_{\mathbf{x} \in \mathcal{B}_t} \cos(\hat{a}_{t,t'}, a_{t'}), \tag{3}$$

where $\mathcal{B}_t = \{\mathbf{x} : |\mathbf{x}| \geq t\}$ contains the audio inputs with length no shorter than $t$.

We empirically compare the averaged cosine similarity at the beginning, middle, and end positions of the speech representations. Figure 2 shows $\bar{s}_{t,t'}$ of the first three ($t' = 1, 2, 3$), middle three ($t' = \lfloor \frac{1+t}{2} \rfloor - 1, \lfloor \frac{1+t}{2} \rfloor, \lfloor \frac{1+t}{2} \rfloor + 1$), and last three ($t' = t - 2, t - 1, t$) positions for each encoding step $t$. At the beginning and middle positions, the averaged cosine similarity $\bar{s}_{t,t'}$ is greater than 0.8 except $t' = 1$, indicating that the representations at such positions in the partial streaming input are close to those in the full speech. Note that $t' = 1$ with a slightly lower similarity won't hurt the performance much, because in practice it is almost impossible to apply *wait*-1 policy in streaming ST. However, the $\bar{s}_{t,t'}$ declines significantly for the end positions, especially for the last one. In addition, we observe that as $t$ becomes larger, the streaming input will gradually approximate the full speech input, then the gap of the speech representation between the offline and the online input becomes smaller. We conclude that **the representations of the end position in the streaming speech are particularly inferior.**

We also average the cosine similarity over both dataset and time dimension with reversed index.

$$\bar{s}_{-t'} = \frac{1}{|\mathcal{B}_{t'}|} \sum_{\mathbf{x} \in \mathcal{B}_{t'}} \frac{1}{|\mathbf{x}| - t' + 1} \sum_{t=t'}^{|\mathbf{x}|} s_{t,t-t'+1}(\mathbf{x}) \tag{4}$$

We calculate the metric $\bar{s}_{-t'}$ of the representations at the last 40 positions in the streaming speech for different methods: Chimera (Han et al., 2021), STEMM (Fang et al., 2022) and MoSST, and

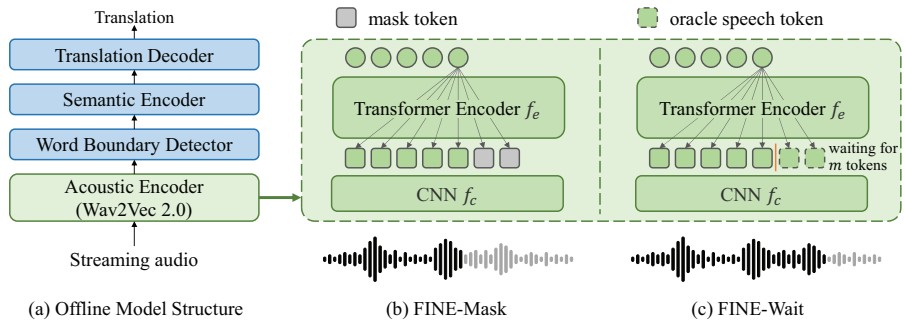

Figure 4: Illustration of offline ST model and proposed methods FINE-Mask and FINE-Wait.

report the results in the Figure 3(a). The consistent results verify that the conclusion above holds and decide that the low-quality representations at the last 10 positions cannot be ignored.

## 3.2 DOES THE POOR REPRESENTATION AT THE LAST POSITIONS OF STREAMING SPEECH AFFECT STREAMING ST PERFORMANCE?

To answer this question, we only calculate the average cosine similarity in the last position for each sample.

$$\forall \mathbf{x}, \quad \bar{s}_{-1}(\mathbf{x}) = \frac{1}{T} \sum_{t=1}^{t=T} \cos(\hat{a}_{t,t}, a_t), \tag{5}$$

$\bar{s}_{-1}(\mathbf{x})$ reflects the degree of deterioration of the representation at the last position of the streaming speech. We sort the dataset by the value of the degree and divide them evenly into 5 groups to ensure enough samples in each group. The translation quality of each group is shown in Figure 3(b). The performance of streaming ST drops close to 10 points as the representation at the last position of the streaming speech becomes worse, while the full-sentence ST fluctuates less than 4 points. In addition, the performance gap between the streaming ST and the full-sentence ST becomes larger as the representation at the last position gets worse. In the worse group, the streaming ST is 12.41 points lower than the full-sentence ST. Therefore, we conclude that **the poor representation at the end position of the streaming speech has a strong effect on the translation quality.**

## 4 FINE: FUTURE-AWARE INFERENCE

Based on these analyses, we find that it is only necessary for the offline ST model to be aware of a short future during streaming encoding. Thus, we propose two **F**uture-aware **IN**ferenc**E** strategies, FINE-Mask and FINE-Wait, to enhance the representations of streaming speech in Figure 4.

### 4.1 FINE-MASK

In this strategy, we use the mask tokens of Wave2Vec2.0 as the pseudo future context and append them to the speech tokens generated from the already consumed speech frames. The mask token embedding is trainable when pre-training Wave2Vec2.0. Particularly, Wav2vec2.0 applies span masks to the speech tokens and reconstructs [2] the corresponding latent features based on unmasked context. By default, the pre-training results in approximately 49% of all time steps being masked with a mean span length of 14.7 (300ms). This pre-training strategy makes the Wav2vec2.0 able to extract better speech representations in offline ST task (Han et al., 2021; Dong et al., 2022; Ye et al., 2022).

Wav2Vec2.0 consists of a multi-layer convolutional subsampler $f_c$ and a Transformer encoder $f_e$. Concretely, for each audio prefix $\hat{\mathbf{x}}_t = (x_1, \ldots, x_t)$ during online inference, the $f_c$ first outputs streaming speech tokens $\hat{\mathbf{c}}_t = (c_1, \ldots, c_\tau)$, where $\hat{\mathbf{c}} \in \mathbb{R}^{\tau \times d}$ and $d$ is the dimension of model and $\tau$ is the sequence length after convolutional subsampling. Then, we concatenate the streaming speech tokens $\hat{\mathbf{c}}$ and $m$ mask token embeddings $\mathbf{e} \in \mathbb{R}^d$ along the time dimension, resulting in a longer

---

[2] Strictly speaking, the task is to identify the quantized latent audio representation rather than reconstruction.

---

**Algorithm 1** Pseudocode of FINE-Mask inference strategy in a PyTorch-like style.

---

```
# model: an offline-trained ST model consists of a acoustic encoder Wav2vec2.0, a token
    boundary detector, a semantic encoder, and a decoder
# m: mask length, K: wait lagging, audio: audio waveform
# mask_emb: pre-trained mask embedding in Wav2vec

N = 0 # the number of source text tokens
x = [] # streaming audio prefix
y = [] # translations
mask_embs = mask_emb.repate(m, 1) # mask embeddings: m × d
while y[-1] != "<eos>":
    if x == audio: # audio has been read
        y = y + model(a,y) # write new target token
    elif N - len(y) < K: # wait K detected source tokens
        x = x + read(audio) # incrementally read audio
        c = model.wav2vec2.cnn(x) # audio tokens τ × d

        # concatenate audio tokens and mask embeddings, (τ + m) × d
        c = torch.cat((c, mask_embs), dim=0)
        a = model.wav2vec2.encoder(c) # audio representations, (τ + m) × d
        a = a[:a.shape[0] - m,:] # discard the predicted representations, τ × d

        if model.token_detector(a): # source text token boundary is detected
            N += 1
    else:
        h = model.semantic_encoder(a)
        y = y + model.decoder(h, y) # write new target token
```

---

sequence of speech tokens $\in \mathbb{R}^{(\tau+m)\times d}$. The new speech tokens are then fed into the Transformer encoder $f_e$, but only the first $\tau$ encoder outputs (i.e., speech features) will be kept for the decoder because, as discussed in Section 3.1, the last $m$ speech features are of poor quality and adversely affect translation quality. The FINE-Mask inference strategy is outlined in Algorithm 1.

### 4.2 FINE-WAIT

In the previous section, we conclude that the speech representations of only the last few positions in the streaming input are inferior. Therefore, a straightforward method is to discard the poor representation at the end positions. In the FINE-Wait strategy, we read more audio frames until the $f_c$ can output $m$ extra speech tokens, resulting in a new sequence of speech features with length $\tau + m$. We then discard the last $m$ speech features as in the FINE-mask strategy above. The FINE-Wait inference strategy is outlined as Algorithm 2 in Appendix. The FINE-Wait strategy incurs additional latency as the model waits for more actual audio frames during each streaming encoding. However, this strategy results in significant improvement, and motivates us to reconsider the trade-off between translation quality and latency. The detailed analysis is given in Section 5.3.

## 5 EXPERIMENTS

### 5.1 EXPERIMENTAL SETTINGS

**Datasets** We evaluate our approach on MuST-C English-German (EnDe), English-Spanish (EnEs) datasets (Di Gangi et al., 2019a). Because limited previous works discussed the MuST-C English-French streaming ST with BLEU-latency curve, we present the EnFr results in Appendix. All the corpora contain source audios, source transcriptions, and target translations, and the results reported are conducted on the corresponding tst-COMMON set. For speech data, we normalize the raw audio wave to the range of $[-1, 1)$. For text data, we keep punctuation and remove non-printing characters, and remain case-sensitive. For each translation direction, the unigram SentencePiece[3] model (Kudo & Richardson, 2018) is used to learn a shared vocabulary of size 10k.

**Model Configuration** We follow MoSST (Dong et al., 2022) to train the offline model. For the acoustic encoder, we use Wav2vec2.0[4] (Baevski et al., 2020) following the base configurations, which use a self-supervised learning framework to pre-train on large-scale audio data from the LibriSpeech (Panayotov et al., 2015) corpus. We use the continuous integrate-and-fire (CIF) module

---

[3] https://github.com/google/sentencepiece
[4] https://dl.fbaipublicfiles.com/fairseq/wav2vec/wav2vec_small.pt

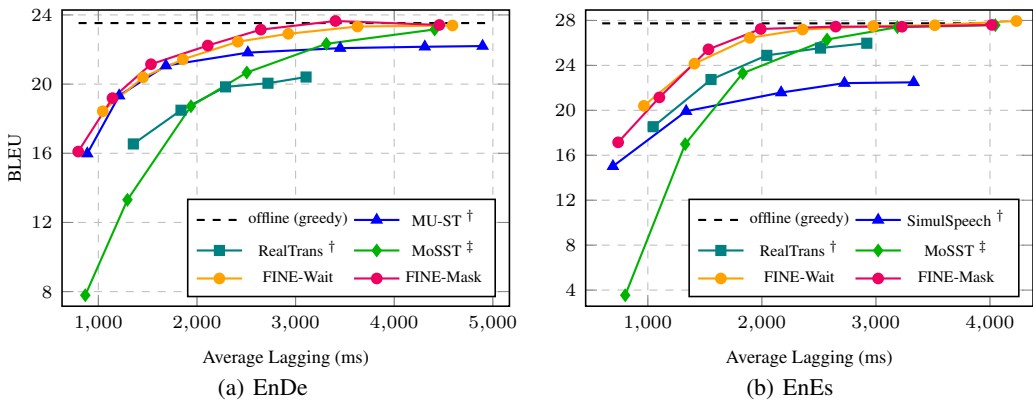

Figure 5: The translation quality (BLEU) against the latency metrics (AL) on the tst-COMMON set of MuST-C EnDe and EnEs dataset. [†] denotes that the results are obtained from corresponding papers. [‡] denotes that the results are from our improved MoSST inference method.

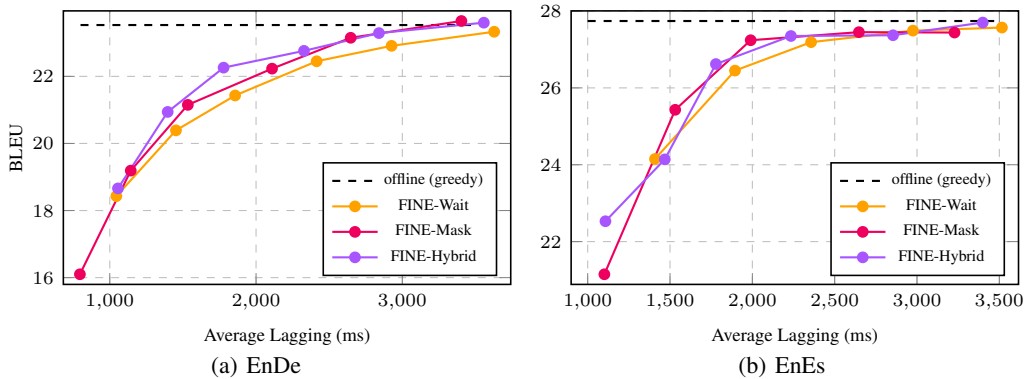

Figure 6: The translation quality (BLEU) against the latency metrics (AL) on the tst-COMMON set of MuST-C EnDe and EnEs dataset.

(Yi et al., 2021) as the word boundary detector. We use 8 and 6 Transformer layers for the semantic encoder and the translation decoder respectively, with 4 attention heads and 768 hidden units.

**Inference** We use the offline-trained ST model to perform streaming-testing with the *wait-k* policy. $k$ here means $k$ detected source text tokens by the CIF word boundary detector. The length of future context tokens ($m$) is 50 and 10 for FINE-Mask and FINE-Wait, respectively. For consistency with previous works (Zeng et al., 2021; Dong et al., 2022), we do not rewrite the tokens that have already been generated during inference. All hyper-parameters are tuned on EnDe and applied to other language pairs. In all experiments, we use our re-implemented MoSST inference method for a better performance (see Appendix B.2 for a detailed explanation).

**Evaluation Metrics** We use SacreBLEU[5] to measure the translation quality. The latency is evaluated with Average Latency (AL) (Ma et al., 2019), Average Proportion (AP) (Cho & Esipova, 2016), and Differentiable Average Lagging (DAL) (Cherry & Foster, 2019) in the SimulEval toolkit[6] (Ma et al., 2020a).

**Baselines** We compare our method with several strong end-to-end streaming ST approaches. (i) **SimulSpeech** (Ren et al., 2020) and **RealTranS** (Zeng et al., 2021) use uni-directional encoder rather than bidirectional to simulate streaming inputs. (ii) **MoSST** (Dong et al., 2022) applies an offline-trained model with a monotonic segmentation module for streaming testing and achieves

---

[5]https://github.com/mjpost/sacrebleu
[6]https://github.com/facebookresearch/SimulEval

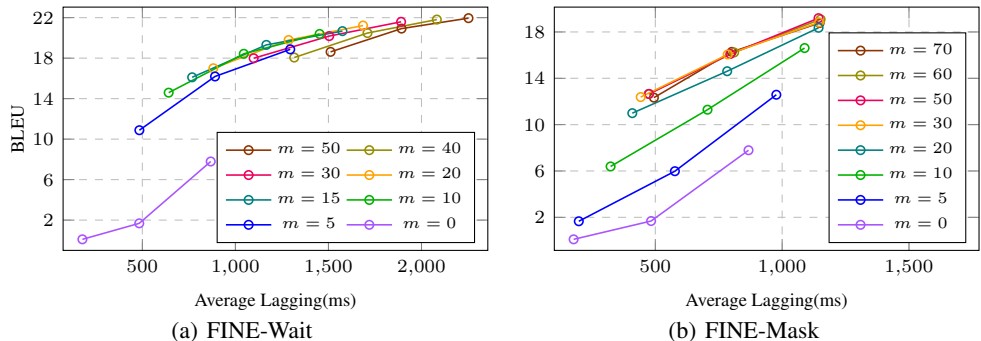

Figure 7: Effect from different lengths of future context. The observed points of FINE-Wait and FINE-Mask in the plots represent wait-$k$ policy with $k = 3, 5, 7$.

competitive performance. For fair comparisons, we use MoSST as our baseline model. (iii) **MU-ST** (Zhang et al., 2022) learns an adaptive segmentation policy to detect meaningful units, which makes read/write decisions.

## 5.2 MAIN RESULTS

We presents the results on the MuST-C EnDe and EnEs tst-COMMON set in Figure 5 [7]. Compared with the online models SimulSpeech and RealTranS, the offline model MoSST achieves higher translation quality with high latency as it encodes bidirectional context information during training. But in the low latency region, it performs poorly one reason for which is the input mismatch between offline-training and online-decoding. With the ability to reduce this mismatch, FINE-Mask and FINE-Wait achieve higher BLEU in all latency regions. In particular, our methods outperform our most compatible base model MoSST by large margins in lower latency regions (when AL is less than 1000$ms$), with improvements over 8 BLEU in EnDe and 12 BLEU in EnEs. This indicates that FINE-Mask and FINE-Wait can effectively mitigate the input mismatch between offline-training and online-decoding. FINE-Wait only requires waiting for a small amount of oracle future context [8], but brings higher translation quality. In addition, our strategies achieve comparable translation quality with full-speech translation at middle latency (at AL around 2000$ms$), especially for EnEs. Compared with FINE-Wait, FINE-Mask achieves a better trade-off as it can introduce more future context without additional latency. Moreover, we find that FINE-Wait and FINE-Mask can be further combined into a new strategy – **FINE-Hybrid**. Specifically, given a streaming speech with length $\tau$, we first wait for $m$ extra oracle speech tokens and add $m'$ mask tokens to the end of the streaming speech. The total length of the streaming speech tokens will be $\tau + m + m'$, but like FINE-Mask and FINE-Wait, only the first $\tau$ encoder outputs will be kept for decoding. We set $m = 10$ and $m' = 50$ for FINE-Hybrid, which are the optimal settings of FINE-Wait and FINE-Mask, respectively. The experimental results are shown in Figure 6. We observe that FINE-Hybrid achieves a better trade-off.

## 5.3 ABLATION STUDY

In this section, we describe experiments to evaluate the effectiveness of our methods from various aspects. All ablation results are obtained from the MuST-C EnDe tst-COMMON set.

### 5.3.1 HOW MUCH FUTURE CONTEXT IS NEEDED?

To answer this question, we compare FINE-Mask and FINE-Wait with different lengths of future context $m$. Figure 7 shows the results. The system that inherits the mismatch problem, i.e. uses the offline model directly for online decoding, is shown by setting $m = 0$. For FINE-Wait, increasing $m$ obtains better translation quality, but generally brings higher latency. Our results in Figure 7(a) shows that it achieves the better trade-off between quality and latency when $m = \{10, 15, 20\}$. If

---

[7]The extended results for other latency metrics (AP and DAL) are described in Appendix B.6.

[8]The results are reported with $m = 10$ for FINE-Wait. It only corresponds to 200$ms$ oracle audio.

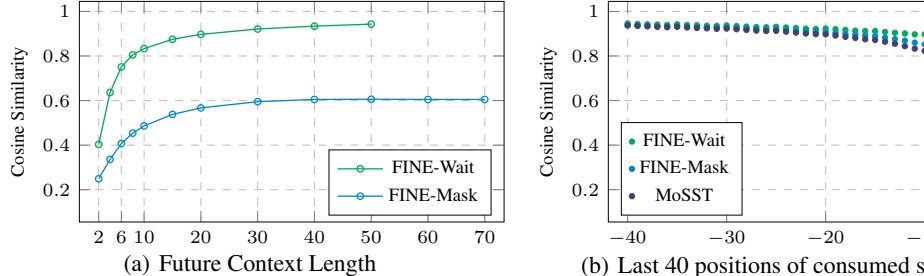

Figure 8: (a). Effect of future context length on the average cosine similarity $\bar{s}_{t,t}$ at the last position of streaming speech. (b). Effect on the average cosine similarity $\bar{s}_{-t'}$ of the forty representation at the end position in the streaming speech. After applying FINE-Mask and FINE-Wait, the representation at the end position is improved.

$m > 20$, the large latency becomes prohibitive. Unlike FINE-Wait, FINE-Mask can attend to more future context without waiting for future audio. It achieves the best trade-off between quality and latency at $m = 50$ (Figure 7(b)). Since the predicted context further into the future will likely introduce more noise, FINE-Mask obtains similar performance when $m$ is increased from 30 to 70. We also investigate the impact of various future context lengths on the representation of the last position by calculating the average cosine similarity in Eq. (4). The results are shown in Figure 8(a). We observe that 1) as $m$ increases, the representation of the last position in the streaming speech becomes better. 2) the curves of the average cosine similarity for FINE-Wait and FINE-Mask becomes flattened when $m > 10$ and $m > 50$, respectively. This is consistent with the results in Figure 7. Therefore, we set future context length $m = 10$ and $m = 50$ respectively for FINE-Wait and FINE-Mask in our other experiments.

### 5.3.2 WHY DOES FINE WORK?

Figure 8(b) plots the changes of average cosine similarity $\bar{s}_{-t'}$ (Eq. (4)) of the last 40 positions in the streaming speech after applying the FINE strategies. FINE-Mask and FINE-Wait achieve at least 0.6 and 0.8 cosine similarity at the last position. In MoSST, however, the cosine similarity is less than 0.6 for the last 4 positions and only 0.2 for the last position. Thus, the speech representations with FINE-Wait and FINE-Mask are closer to those of the full speech input, especially at the end positions. The cosine similarities in FINE-Mask are lower than those in FINE-Wait because the predicted future context is less accurate by consuming mask tokens. However, FINE-Mask achieves better balance (Figure 5) as it does not incur additional latency. In sum, introducing future context significantly reduces the representation gap between full and partial speech input, improves streaming speech representations, and achieves a better balance between quality and latency.

More ablation studies are included in the Appendix. In Appendix B.3 we find the least monotonic examples are mostly improved by our strategies. Appendix B.4 analyzes the difference of the pre-trained and fine-tuned Wav2Vec2.0 with respect to future context representations. Appendix B.5 demonstrates why all predicted features should be discarded in FINE-Mask.

## 6 CONCLUSION

In this paper, we examine streaming speech translation from a new perspective. We investigate the effects of the input mismatch between offline-training and online-decoding. We find that the representations at the end positions in the streaming input are particularly poor, directly impacting the translation quality. We propose FINE-Mask and FINE-Wait to improve these representations by introducing, respectively, predicted and real future context. Experiments and analysis demonstrate their effectiveness in bridging the representation gap between full speech encoding and partial streaming encoding. Furthermore, our strategies can be generally beneficial to streaming speech translation models that are based on Wav2Vec2.0. In the future, we will experiment with other methods to improve the accuracy of predicting future information. We hope the work and perspective presented in this paper can engender further innovations in general streaming translation.

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

---

**Algorithm 2** Pseudocode of FINE-Wait inference strategy in a PyTorch-like style.

```
# model: an offline-trained ST model consists of a acoustic encoder Wav2vec2.0, a token
    boundary detector, a semantic encoder, and a decoder
# m: mask length, K: wait lagging, audio: audio waveform

N = 0 # the number of source text tokens
x = [] # streaming audio prefix
y = [] # translations
while y[-1] != "<eos>":
    if x == audio: # audio has been read
        y = y + model(a,y) # write new target token
    elif N - len(y) < K: # wait k detected source tokens
        x = x + read(audio) # incrementally read audio
        c = model.wav2vec2.cnn(x) # audio tokens
        if c.shape[0] <= m: # waiting for m audio tokens
            continue
        a = model.wav2vec2.encoder(c) # audio representations, (τ + m) × d
        a = a[:a.shape[0] - m,:] # discard last m representations, τ × d

        if model.token_detector(a): # source text token boundary is detected
            N += 1
    else:
        h = model.semantic_encoder(a)
        y = y + model.decoder(h, y) # write new target token
```

---

# A    ALGORITHM OF FINE-WAIT

The pseudo code of FINE-Wait are described in Algorithm 2

# B    ADDITIONAL EXPERIMENTS

## B.1    WHY WE USE AL RATHER THAN $k$?

In our presented results, we plot the BLEU *v.s.* AL rather than $k$. We argue that $k$ is not a fair metric to evaluate the latency. In text streaming translation, different tokenization (*e.g.*, different number of BPE operations) will lead to different token boundaries for the same sentence. It indicates the $k$ tokens do not necessarily represent the same partial sentence for different BPE methods. This situation becomes even severer for speech streaming translation. As we have a source text token boundary detector in our model, the first $k$ detected text tokens will represent different lengths of audio frames for different input audios. To be precise, the wait-$k$ policy used in our streaming speech translation is actually wait-$k$ detected tokens policy. Therefore, we prefer to use AL rather than $k$ as the latency metric in our experiments.

## B.2    WHY WE USED OUR IMPLEMENTED MOSST?

For streaming speech translation, when the allowed AL is increasing, the performance of streaming ST will gradually converge to the offline model. However, in the original implementation of MoSST, there is still a nonnegligible gap between the SST and offline ST for the large AL. We re-implement the MoSST inference method and build our inference strategies on top of the improved MoSST inference. The difference of the inference performance between the original MoSST and our improved MoSST can be seen from Figure 9. We can see the inference BLEU of our improved MoSST can approximate the offline model as the AL increases. The detailed implementation can refer to the submitted code.

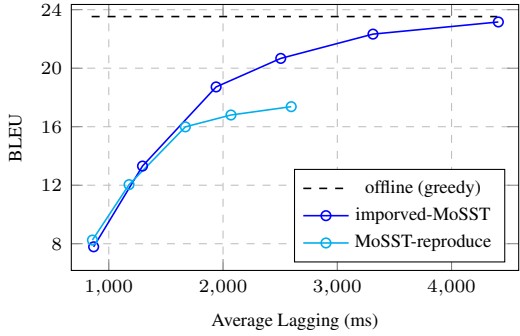

Figure 9: Difference between original and improved MoSST.

Table 1: Performance (BLEU) on different monotonic levels on test set of MuST-C EnDe.

| Monotonic Level | **Easy** | **Medium** | **Hard** | **AL** |
|---|---|---|---|---|
| # example | 668 | 1013 | 895 | - |
| Offline (greedy) | 29.36 | 23.32 | 22.38 | - |
| MoSST | 24.29 | 15.32 | 9.22 | 1295 |
| FINE-Wait | $26.61^{+2.32}$ | $19.36^{+4.04}$ | $17.24^{+8.02}$ | 1251 |
| FINE-Mask | $26.03^{+1.74}$ | $19.35^{+4.03}$ | $\mathbf{17.40^{+8.18}}$ | 1143 |

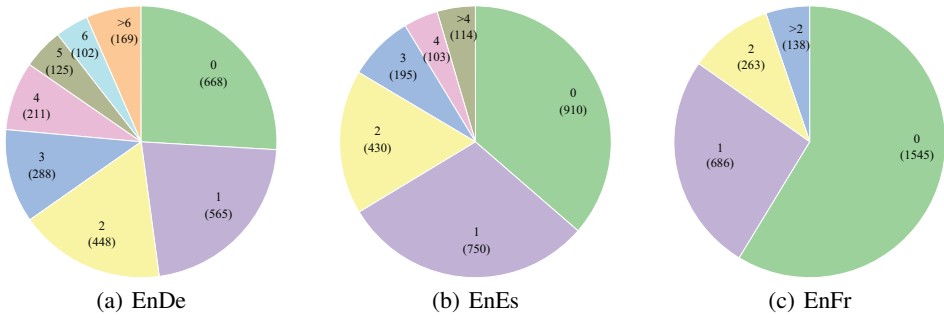

(a) EnDe        (b) EnEs        (c) EnFr

Figure 10: The source-to-target alignment position shift on MuST-C EnDe, EnEs, and EnFr tst-COMMON set.

### B.3 WHAT EXAMPLES ARE IMPROVED BY OUR STRATEGIES?

For tst-COMMON on MuST-C EnDe, we use fast-align[9] to identify the token-level alignment between source transcription and target translation following Zhang & Feng (2022c). First, we define the source-to-target alignment position shift as $\max\{0, i - j\}$, where the $i$th source token is aligned to the $j$th target token. If $i - j$ is large, it means in order to translate the $j$th target token, the model may need to read more until seeing the $i$th source token. Then we calculate the monotonic level of each example as the averaged alignment position shift over the number of aligned tokens, *i.e.*,

$$\text{monotonic\_level} = \frac{1}{|\text{aligned\_pairs}|} \sum_{(i,j) \in \text{aligned\_pairs}} \max\{0, i - j\} \qquad (6)$$

We divide the test set into 3 groups according to different monotonic levels: easy ($= 0$), medium ($< 3$) and hard ($\geq 3$). For each group, we evaluate different inference methods and report the results in Table 1. As we explained in B.1, it is almost impossible to guarantee the same AL for different inference methods. For a fair comparison, we try our best to set the AL of different methods to be approximately equal. We can see our inference strategies show a significant advantage on the non-monotonic examples (hard group).

### B.4 HOW IMPORTANT OF THE WAV2VEC2.0?

As we mentioned in the main text, the special audio token "mask" in Wav2Vec2.0 is pre-trained on the Librispeech dataset to reconstruct the corresponding feature conditional on unmasked context via the contrastive task. In our experiments, we didn't include contrastive learning as the auxiliary task in the downstream ST training. And in our FINE-Mask inference, we directly leverage the mask embeddings as the future context by appending them to the streaming input. However, we found the speech representations after ST training becomes even better. Particularly, we calculate the cosine similarity between every predicted future representation and full speech representations at the same position, and the results are illustrated in Figure 11. On either the Librispeech or the MuST-C audio test set, the fine-tuned Wav2Vec2.0 can produce better speech representations from the masking inputs.

---

[9] https://github.com/clab/fast_align

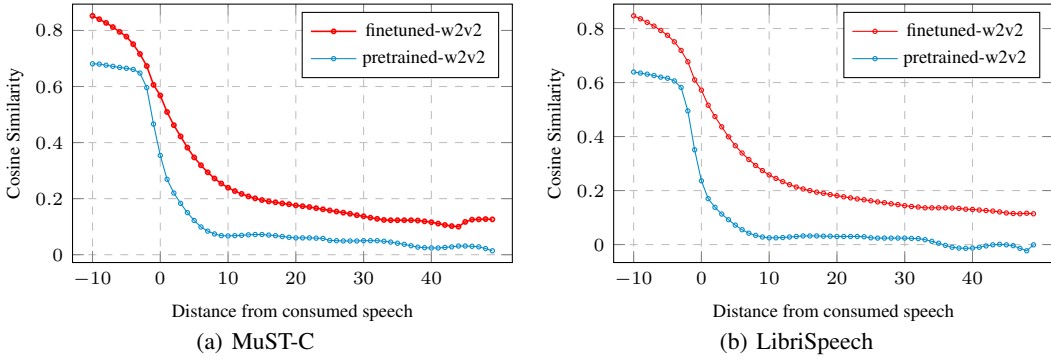

(a) MuST-C            (b) LibriSpeech

Figure 11: We measure the accuracy of predicted context by calculating the cosine similarity between every predicted future representation and full speech representations at the same position.

## B.5 WHY ARE ALL PREDICTED FEATURES DISCARDED?

In FINE-Mask strategy, all the output representations corresponding to the $m = 50$ masking tokens will be discarded, because we have demonstrated that the representations at the ending positions are inferior. However, as shown in 11, the first 10 predicted representations are not as bad as the next 40. Therefore, on the EnDE test set, we also conduct another streaming ST inference by appending different numbers of predicted context to the original speech representations. We use discard rate $p$ to measure the number of appending features. When $p = 1.0$, all predicted features are discarded and it reduces to the standard FINE-Mask inference. In Figure 12, we compare the streaming speech translation qual-

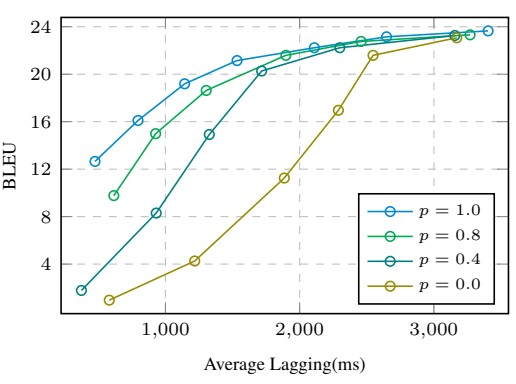

Figure 12: BLEU *v.s.* AL on different $p$.

ity between regular FINE-Mask and its variant. It is concluded that the predicted future context is too noisy and harmful to the performance.

## B.6 ADDITIONAL RESULTS ON ENDE/ES AND ENFR

In this section, we evaluate our inference methods with other latency metrics AP and DAL. The AP-BLEU and DAL-BLEU curves on the MuST-C EnDe, EnEs, and EnFr tst-COMMON sets are shown in Figure 14. For both EnDe and EnEs, our proposed inference strategies can consistently improve the baseline by a large margin.

**EnFr** In general, we found very limited previous works evaluating on the MuST-C EnFr dataset with BLEU-latency curves. We apply our strategies on the EnFr datasets on different latency metrics and present the results in Figure 13 and Figure 14(e) and 14(f). The FINE-Mask strategy can achieve significant improvement. Note that the $y$-axis of EnFr has a larger measuring scale than EnDe or EnEs, so the gap

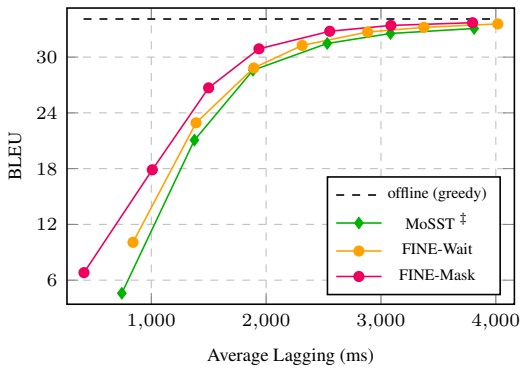

Figure 13: BLEU *v.s.* AL on the tst-COMMON set of MuST-C EnFr dataset.

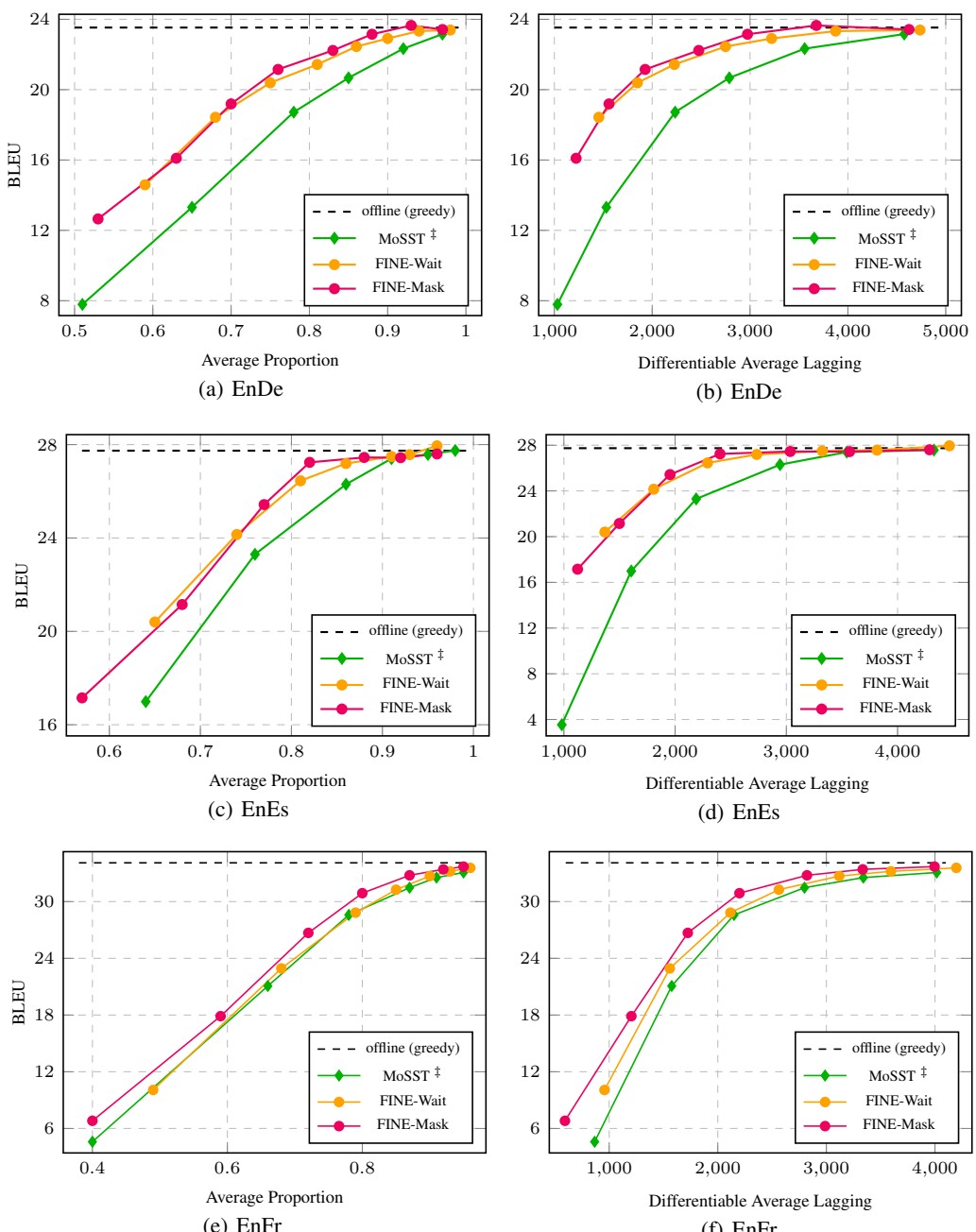

Figure 14: The translation quality (BLEU) against the latency metrics (AP, DAL) on the tst-COMMON set of MuST-C EnDe, EnEs and EnFr dataset. ‡ denotes that the results are from our improved MoSST.

is not as visible as the other two language pairs, *e.g.,* the absolute BLEU gain on AL (1000ms) from MoSST to FINE-Mask is about 6 BLEUs. We observe that FINE-Wait does not bring as significant improvement over the baseline improved MoSST. We also count the distributions of monotonic levels of EnFr test set as shown in Figure 10(c), and find the majority (94.8%) of the averaged position shift is < 3. Most of the EnFr examples fall into the easy and medium groups. According to the finding in Table 1, it should be common that there is a less significant difference between our strategy and the baseline MoSST.

### B.7 How Do Fine-tuning Epochs Affect the FINE-Mask Strategy?

For FINE-Mask strategy, we use a mask token that is well-trained during pre-training of the Wav2Vec 2.0 model to predict future speech features. Now we investigate the performance of different fine-tuning steps on FINE-Mask strategy. We evaluate the FINE-Mask inference and regular streaming inference (MoSST) at different checkpoints with epoch $e = \{23, 25, 27, 29, 31\}$ after training converges, as illustrated in Figure 15. Although different checkpoints achieve different performances, FINE-Mask consistently brings improvement (about 6 BLEU at low latency regimes) over the baseline. Moreover, we can observe that the performance gap between different checkpoints in MoSST is larger than that in FINE-Mask. Specifically, when AL $\in [1000, 1500]$, the performance gap between the best and the worst checkpoints in MoSST is 3 BLEU, while the gap in FINE-Mask is only 2 BLEU. All these indicate that the number of fine-tuning steps is not sensitive to the performance improvement of FINE-Mask.

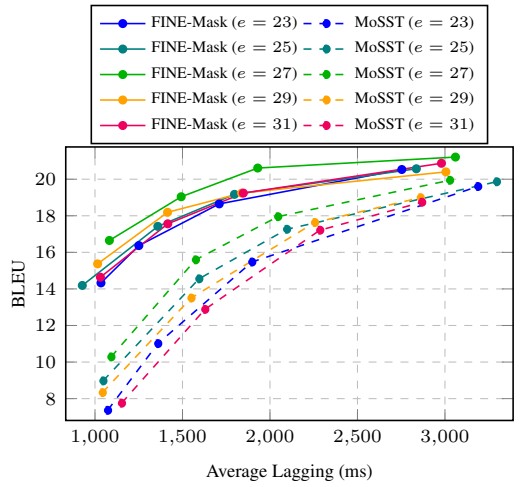

Figure 15: Performance of different fine-tuning epochs on FINE-Mask strategy.

### B.8 Future Work: Upper Bounder of FINE-Mask

If we infer with FINE-Wait without counting on the latency, the result will be the upper bound of the FINE-Mask, as illustrated in Figure 16. We can observe that the translation quality is further improved at all latency regimes when the accuracy of the predicted contexts becomes fully correct. In particular, the translation quality remains 16 BLEU at a very low latency regime (AL is about only 200$ms$), exceeding the FINE-Mask by about 10 BLEU. Thus it motivates us to predict the future contexts or representations as accurate as possible and we will explore this direction in the future work.

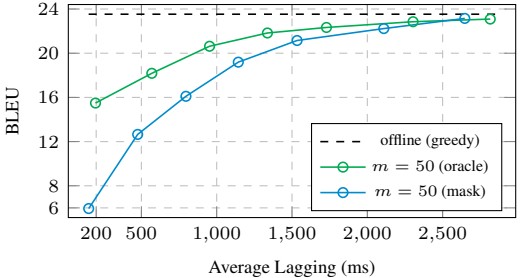

Figure 16: Upper Bound of FINE-Mask.

## C Differences from Streaming Speech Recognition techniques

We briefly describe the differences between our work and other streaming speech processing techniques for other readers from other backgrounds to understand our work. Most streaming recognition techniques still fall into the category to train the model in a streaming manner, e.g., MoChA (Chiu & Raffel, 2018). Similarly, CAAT (Liu et al., 2021) follows the RNN-T (Graves, 2012) method to build a streaming speech translation model. However, what we are researching is completely different from these streaming recognition techniques. We want to reuse the pre-trained offline ST model for streaming inference without further streaming-related training. In general, this is a difficult problem for two reasons. First, the streaming inference takes the partial speech sequence as input, while the offline model always consumes the full speech sequence. Second, the reordering issue prevents partial speech sequence from seeing future information. Our proposed method attempts to relieve such mismatches via the FINE-Mask strategy.

Moreover, the word boundary detector in our model is similar to that in SN model (Wang et al., 2020b) However, there are two significant differences. First, in our pseudo word detector, we use CIF module. For detector architecture, SN and CIF are almost identical with a linear layer and the sigmoid activation. The only difference is the label used. The label for CIF is the total number of words in the transcription. During inference, the CIF module will segment the speech sequence

into several possibly overlapped segments, whose number should be roughly equal to the number of words if the model is well-trained. However, for SN, it uses actual word-frames alignment information as the labels to learn the real word boundary. So CIF is almost unsupervised or weakly supervised learning, while SN is supervised learning. Second, our methods are mainly used to adapt a single pre-trained offline ST model to streaming inference without further streaming training, and alleviate the input mismatch between offline training and streaming inference. However, for SN, it is trained conditioned on the previous states. It indicates SN is a streaming-trained model with a unidirectional encoder.

In summary, our method is different from traditional streaming techniques.

## D    WHY MoSST AS A BASE MODEL

MoSST (Dong et al., 2022) consists of an acoustic encoder, a monotonic segmentation module (*i.e.*, CIF module), a semantic encoder, and a translation decoder. First, the acoustic encoder is pre-trained Wav2Vec2.0. The mask speech embedding is well-trained and is helpful for our FINE-Mask strategy. Moreover, the CIF module can be trained to roughly segment the speech sequence into words level with very weak signal (the total number of words). After training, the number of resulted segments is roughly equal to the number of words. So it can be used as a pseudo word detector for streaming inference. Thus, we use MoSST as our base model.

**Details of Offline Training** The offline ST model is first trained for about 20 epochs by a multi-task learning, including ASR and ST tasks. A language identity tag is prepended to the target sentence for indicating which task is learned. In this stage, the CIF module which is used for monotonic segmentation is deactivated, in other words, the CIF module is not trained. The main purpose is to learn a better decoder, i.e., a well-trained language model. Then, we activate the CIF module such that its parameters are trainable, and continue to train for another several epochs. In this stage, only the ST task is learned. Usually, it will only take 5 or 6 epochs to converge.

## E    NUMERIC RESULTS FOR THE FIGURES

Table 2: Numeric results for Figure 5, 13, 14.

| Model | EnDe | | | | EnEs | | | | EnFr | | | |
|---|---|---|---|---|---|---|---|---|---|---|---|---|
| | AL | AP | DAL | BLEU | AL | AP | DAL | BLEU | AL | AP | DAL | BLEU |
| MoSST | 867 | 0.51 | 1032 | 7.79 | 802 | 0.38 | 981 | 3.54 | 743 | 0.4 | 866 | 4.59 |
| | 1295 | 0.65 | 1531 | 13.31 | 1327 | 0.64 | 1606 | 16.99 | 1376 | 0.66 | 1576 | 21.07 |
| | 1939 | 0.78 | 2234 | 18.72 | 1832 | 0.76 | 2189 | 23.30 | 1886 | 0.78 | 2150 | 28.59 |
| | 2505 | 0.85 | 2788 | 20.67 | 2571 | 0.86 | 2944 | 26.30 | 2532 | 0.87 | 2798 | 31.47 |
| | 3312 | 0.92 | 3559 | 22.33 | 3188 | 0.91 | 3543 | 27.40 | 3083 | 0.91 | 3341 | 32.54 |
| | 4410 | 0.97 | 4576 | 23.16 | 4047 | 0.95 | 4327 | 27.58 | 3811 | 0.95 | 4017 | 33.08 |
| FINE-Mask | 796 | 0.63 | 1223 | 16.10 | 740 | 0.57 | 1123 | 17.15 | 414 | 0.4 | 593 | 6.81 |
| | 1143 | 0.70 | 1559 | 19.19 | 1101 | 0.68 | 1499 | 21.15 | 1009 | 0.59 | 1205 | 17.88 |
| | 1534 | 0.76 | 1928 | 21.15 | 1532 | 0.77 | 1954 | 25.43 | 1499 | 0.72 | 1724 | 26.69 |
| | 2109 | 0.83 | 2476 | 22.23 | 1990 | 0.82 | 2404 | 27.24 | 1937 | 0.8 | 2200 | 30.89 |
| | 2647 | 0.88 | 2974 | 23.15 | 2647 | 0.88 | 3035 | 27.45 | 2554 | 0.87 | 2821 | 32.78 |
| | 3404 | 0.93 | 3678 | 23.65 | 3228 | 0.92 | 3568 | 27.44 | 3087 | 0.92 | 3335 | 33.41 |
| | 4457 | 0.97 | 4625 | 23.42 | 4017 | 0.96 | 4287 | 27.60 | 3797 | 0.95 | 3996 | 33.70 |
| FINE-Wait | 1045 | 0.68 | 1455 | 18.43 | 965 | 0.65 | 1368 | 20.40 | 841 | 0.49 | 957 | 10.07 |
| | 1453 | 0.75 | 1849 | 20.39 | 1409 | 0.74 | 1808 | 24.15 | 1390 | 0.68 | 1558 | 22.93 |
| | 1857 | 0.81 | 2227 | 21.43 | 1895 | 0.81 | 2291 | 26.45 | 1892 | 0.79 | 2116 | 28.83 |
| | 2414 | 0.86 | 2750 | 22.45 | 2357 | 0.86 | 2734 | 27.19 | 2314 | 0.85 | 2563 | 31.26 |
| | 2926 | 0.90 | 3220 | 22.91 | 2975 | 0.91 | 3323 | 27.49 | 2884 | 0.9 | 3119 | 32.70 |
| | 3628 | 0.94 | 3875 | 23.33 | 3516 | 0.93 | 3815 | 27.57 | 3374 | 0.93 | 3594 | 33.20 |
| | 4590 | 0.98 | 4737 | 23.39 | 4232 | 0.96 | 4466 | 27.95 | 4017 | 0.96 | 4197 | 33.57 |

Table 3: Numeric results for Figure 5, 6. The results of **MU-ST** are obtained from (Zhang et al., 2022). The results of **SimulSpeech** and **RealTrans** are obtained from (Zeng et al., 2021).

| | | | | | | | | | |
|---|---|---|---|---|---|---|---|---|---|
| | **MU-ST** | | | | | | | | |
| | AL | 1023 | 1424 | 1953 | 2642 | 3621 | 4453 | 5089 | 5754 |
| | BLEU | 17.94 | 20.85 | 22.78 | 24.3 | 24.82 | 24.99 | 25.05 | 25.9 |
| EnDe | **RealTrans** | | | | | | | | |
| | AL | 1355 | 1838 | 2290 | 2720 | 3106 | - | - | - |
| | BLEU | 16.54 | 18.49 | 19.84 | 20.05 | 20.41 | - | - | - |
| | **FINE-Hybrid** | | | | | | | | |
| | AL | 1056 | 1396 | 1778 | 2328 | 2840 | 3557 | - | - |
| | BLEU | 18.66 | 20.94 | 22.26 | 22.76 | 23.29 | 23.60 | - | - |
| | **SimulSpeech** | | | | | | | | |
| | AL | 694 | 1336 | 2169 | 2724 | 3331 | - | - | - |
| | BLEU | 15.02 | 19.92 | 21.58 | 22.42 | 22.49 | - | - | - |
| EnEs | **RealTrans** | | | | | | | | |
| | AL | 1047 | 1554 | 2043 | 2514 | 2920 | - | - | - |
| | BLEU | 18.54 | 22.74 | 24.89 | 25.54 | 25.97 | - | - | - |
| | **FINE-Hybrid** | | | | | | | | |
| | AL | 1108 | 1468 | 1779 | 2234 | 2854 | 3399 | - | - |
| | BLEU | 22.53 | 24.14 | 26.62 | 27.35 | 27.37 | 27.70 | - | - |

Table 4: Numeric results for Figure 7.

| **FINE-Mask** | | | | | | **FINE-Wait** | | | | | |
|---|---|---|---|---|---|---|---|---|---|---|---|
| $m$ | AL | BLEU | $m$ | AL | BLEU | $m$ | AL | BLEU | $m$ | AL | BLEU |
| 0 | 178 | 0.10 | 30 | 442 | 12.38 | 0 | 178 | 0.10 | 20 | 882 | 17.00 |
| | 483 | 1.68 | | 785 | 16.06 | | 483 | 1.68 | | 1286 | 19.79 |
| | 867 | 7.79 | | 1146 | 18.83 | | 867 | 7.79 | | 1686 | 21.22 |
| 5 | 198 | 1.66 | 50 | 475 | 12.65 | 5 | 484 | 10.88 | 30 | 1098 | 17.99 |
| | 577 | 5.98 | | 796 | 16.10 | | 891 | 16.19 | | 1504 | 20.18 |
| | 977 | 12.58 | | 1143 | 19.19 | | 1295 | 18.87 | | 1890 | 21.59 |
| 10 | 324 | 6.39 | 60 | 473 | 12.65 | 10 | 641 | 14.59 | 40 | 1316 | 18.06 |
| | 706 | 11.29 | | 811 | 16.23 | | 1045 | 18.43 | | 1710 | 20.48 |
| | 1088 | 16.61 | | 1152 | 19.09 | | 1453 | 20.39 | | 2082 | 21.81 |
| 20 | 409 | 10.99 | 70 | 494 | 12.33 | 15 | 768 | 16.11 | 50 | 1511 | 18.62 |
| | 783 | 14.61 | | 801 | 16.28 | | 1166 | 19.31 | | 1893 | 20.92 |
| | 1144 | 18.36 | | 1148 | 18.77 | | 1575 | 20.67 | | 2253 | 21.95 |

We also provide the numeric results for Figures 5, 13, and 14 in Tables 2, for Figure 7 in Table 3, and for Figures 5 and 6 in Table 4.

