# OpenReview forum: "FINE: Future-Aware Inference for Streaming Speech Translation"
_ICLR.cc/2023/Conference — Submitted to ICLR 2023_

### Official Review · Reviewer_JSnA · 2022-10-25

**Confidence:** 4
**Correctness:** 4
**Technical Novelty And Significance:** 3
**Empirical Novelty And Significance:** 3
**Recommendation:** 6

**Clarity, Quality, Novelty And Reproducibility:**

Clarity: The paper is clearly written.

Quality: Standard.

Novelty: Marginal. The novelty is mainly contributed by the FINE-mask method. For the FINE-wait method, it is not new and is just slightly modified from existing wait-k method.

Reproducibility: The codes are provided in the paper and others can replicate the results.



**Strength And Weaknesses:**

Strength:
The paper propose two different kinds of method and good results are obtained.


Weaknesses:
Although the investigation may draw a lot of interest, the conclusion is not surprising.
In 3.1, the investigation shows that the representations at the end position has a higher gap with the full input case. This is not surprising as   frames near the boundary has no future context comparing with the full input case.
By the way, if the end position is actually the end position of the whole sentence, the representations should have no difference with the full input case.

I also think the authors should compare their methods with the idea in "Low latency end-to-end streaming speech recognition with a scout network"

The FINE-wait method proposed by the paper is not new.

**Summary Of The Paper:**

This paper investigates the streaming property in ST and propose new streaming methods.
The two proposed methods: FINE-mask and FINE-wait are proven to be effective and achieve better trade-offs between translation quality and latency.
Furthermore, these methods can effectively alleviate the mismatch problem between offline training and online inference.

**Summary Of The Review:**

This paper is motived by an observation that representations at the end position are usually very different from the full input case.
The authors try to solve this problem by proposing two different methods and good results are obtained.

---

> ### Author Response · Authors · 2022-11-18
> **Thanks for the insightful comments. Here is the response to Reviewer JSnA**
>
> Thanks for the insightful comments.
>
> We would like to address the main contribution. We are not working on the traditional research direction of streaming ST -- streaming model training, but another research direction: how to adapt a single pre-trained offline ST model to streaming inference without further streaming training, and how to close the gap between offline pre-training and online inference. \
> However, most prior works on streaming ST try to tackle the streaming inference via training in a streaming  fashion. Particularly, if some methods require the fixed k or N (in wait-k or stride-N) during training, the inference should also use the same k or N. To meet the requirement of different inference latency, multiple ST models are trained in different k and N. But we only need one single offline ST model.
>
> > Q1: Although the investigation may draw a lot of interest, the conclusion is not surprising. In 3.1, the investigation shows that the representations at the end position has a higher gap with the full input case. This is not surprising as frames near the boundary has no future context comparing with the full input case.
>
> A1: Yes, we agree your intuition on the quality gap at the end position between streaming input and full input. In Sec 3.1, we want to quantify this intuition and make it rigorous. Actually, we found each frame representation of streaming speech input becomes poor, but the representations of the end positions (about last 0 positions) in the streaming speech are particularly inferior. This conclusion motivates us to propose FINE-Mask and FINE-Wait and provides an evidence to select the number of masking and future speech tokens in two strategies. We believe that this analysis can constitute an interesting avenue for future research on reusing the offline ST model for streaming inference.
>
> > Q2: By the way, if the end position is actually the end position of the whole sentence, the representations should have no difference with the full input case.
>
> A2: The end position we are discussed in Sec 3.1 is the end position of the streaming input at every time stamp. Only for the streaming input at the last time stamp, it is equivalent to the end position of the whole sentence.
>
> > Q3: the authors should compare their methods with the idea in "Low latency end-to-end streaming speech recognition with a scout network".
>
> A3: Thanks for pointing this reference with the similar idea for word boundary detector. However, there are two significant differences between the idea in "scout network" (SN) and our approach. \
> First, in our **pseudo** word detector, we use CIF module. For detector architecture, SN and CIF are almost identical with a linear layer and the sigmoid activation. The only difference is the label used. The label for CIF is the total number of words in the transcription. During inference, the CIF module will segment the speech sequence into several possibly overlapped segments, whose number should be roughly equal to the number of words if the model is well-trained. However, for SN, it uses actual word-frames alignment information as the labels to learn the **real** word boundary. So CIF is almost unsupervised or weakly supervised learning, while SN is supervised learning. \
> Second, our methods are mainly used to adapt a single pre-trained offline ST model to streaming inference without further streaming training, and alleviate the input mismatch between offline training and streaming inference. However, for SN, it is trained conditioned on the previous states. It indicates SN is a streaming-trained model with a unidirectional encoder. So it is difficult to directly compare them in performance. \
> We add the discussion of SN in the Appendix C.

---

### Official Review · Reviewer_3NNH · 2022-10-26

**Confidence:** 3
**Correctness:** 4
**Technical Novelty And Significance:** 4
**Empirical Novelty And Significance:** 4
**Recommendation:** 8

**Clarity, Quality, Novelty And Reproducibility:**

The paper is clearly written and detailed. The experiments are thorough and detailed and the ideas are simple and novel. Code and work should be reproducible.

**Strength And Weaknesses:**

### Strengths
- Both FINE mask and FINE wait methods are simple modifications for streaming speech translation
- FINE Mask is a clever strategy to make the output representations similar to offline full context speech and streaming speech.
- Results compared to MoSST baseline are quite significant in the low latency region.
- Paper is well written and supports the hypothesis with extensive experiments and analysis.


### Questions
- FINE-wait approach is not entirely novel, and is also mostly known as a latency/performance trade-off, is there anything different that I missed?
- As I understand the authors reimplement the MOSST baseline. From Fig 5 RealTrans and MU-ST perform better on low latency region. Can FINE mask and FINE wait extended to other existing methods that perform better in the low latency region?
- Recently Conformer architecture has shown improvements over transformer for speech related tasks. Did the authors consider trying better encoder layers like conformer as well as better pretraining methods for the encoder like w2vBERT? Will that make applying FINE-mask approach difficult?


**Summary Of The Paper:**

The paper investigates the difference is speech representations towards end for streaming input vs when complete utterance input is available. Based on this investigation the work proposes future aware streaming ST which introduces few frames of future speech signals. Extensive experiments verify the improvements compared to other recent works.

**Summary Of The Review:**

Overall the work is novel and interesting, well supported by experiments and analysis. Hence my high rating.

---

> ### Author Response · Authors · 2022-11-18
> **Thanks for the insightful and encouraging comments. Here is the response to additional questions for Reviewer 3NNH.**
>
> > Q1: FINE-wait approach is also mostly known as a latency/performance trade-off
>
> A1: Yes, you're right. It's worth mentioning that FINE-Wait only uses very short future information (200ms oracle audio), but achieves amazing performance improvement in the low-latency region. The different part is that we didn't directly apply this strategy on streaming trained ST model, and we found it works on the scenario that a pre-trained offline ST model can be re-used as streaming inference without any additional streaming training.
>
> However, FINE-Wait didn't trivially work well on any similar case. For example, we also have an experiment to test an offline NMT model with FINE-Wait inference. With the same wait-k policy, FINE-Wait inference by peeking 1 or 2 tokens can achieve a significant BLEU improvement over regular inference. But when considering the AL, the BLEU vs AL curve of two inference methods are highly coincident. In other words, the BLEU improvement by FINE-Wait is cancelled by the increased latency, even only 1 or 2 tokens are peeked. Therefore, how to apply FINE-Wait strategy on general offline->online inference scenario should be also an interesting research topic.
>
> > Q2: Can FINE mask and FINE wait extended to other existing methods (RealTrans, MU-ST) that perform better in the low latency region?
>
> A2: Overall speaking, our proposed strategies are expected to alleviate the mismatch between offline-trained model and streaming inference. These methods are already trained in streaming fashion. We're not sure whether they still need to solve such issue. \
> Technically, it is difficult to directly extend FINE-Mask to other methods, because at least two important conditions are required. First, the acoustic model should be pre-trained Wav2Vec2.0, where mask speech tokens are well-trained and helpful in FINE-Mask. Second, a continuous integrate-and-fire (CIF) module is trained to roughly segment the speech sequence into words level with very weak signal (the total number of words). After training, the number of resulted segments is roughly equal to the number of words. So it can be used as a pseudo word detector for streaming inference. For RealTrans, it didn't use Wav2Vec2.0 as acoustic encoder, but its CTC module could possibly be used a word detector. For MU-ST, it already uses Wav2Vec2.0 and a multi-modal meaning full unit detector. However, unlike a simple linear layer as CIF, the MU module involving audio and target is complicated and multi-modal. How to reasonably integrate mask speech token into MU is the main problem. \
> In contrast, we think it is more possible to apply FINE-Wait inference for other methods. However, we still need to figure out the appropriate layer to discard low quality speech features at ending positions. If these methods (RealTrans, MU-ST) are open-sourced, we would like to try.
>
> > Q3: Did the authors consider trying better encoder layers like conformer as well as better pre-training methods for the encoder like w2vBERT? Will that make applying FINE-mask approach difficult?
>
> A3: Yes, we agree better pre-trained acoustic encoder will make FINE-Mask better, especially, when the masking tokens are well trained to recover the original speech features. We currently adopt Wav2Vec2.0 base model because it has a relatively smaller model size than w2vBERT. But we agree it is a great future work to explore the better model architectures and pretraining objectives, like a smaller-sized w2vBERT. Thanks for the suggestion.

---

### Official Review · Reviewer_2wmu · 2022-10-27

**Confidence:** 3
**Correctness:** 3
**Technical Novelty And Significance:** 2
**Empirical Novelty And Significance:** 3
**Recommendation:** 5

**Clarity, Quality, Novelty And Reproducibility:**

This paper is clearly written overall, and it is reproducible. It has limited originality.

**Strength And Weaknesses:**

Strength: The proposed methodology is technically sound. Overall, this paper clearly presents the proposed approach, with well designed analysis and experiments presented in both main paper and appendix.

Weakness: The proposed FINE-Mask and FINE-Wait are mostly based on existing technics, with limited novelty. For example, FINE-Wait appears to be a standard practice to incorporate future contexts. Also I think the experimental section may need clarification on several items (see questions in the summary section below).

**Summary Of The Paper:**

In this paper authors present their work on analyzing speech representation mismatch between non-streaming and streaming setting, for speech translation. Based on it, authors propose to leverage future-aware inference (FINE) (including FINE-Mask and FINE-Wait) for mitigation, and experimental results show the effectiveness of the proposed approach.

**Summary Of The Review:**

This paper clearly presents authors' work to improve streaming speech translation, including mismatch analysis, the proposed FINE-Mask and FINE-Wait approaches, experimental design and ablation studies etc. While there are good thoughts and experiments put in it, I find the proposed approach is mostly based on existing techniques or standard practice. I'd also suggest authors consider addressing the following questions:

1. For the results shown in Figure 5, do authors have comments on model size and computation cost, when comparing the proposed methods vs. baselines?
2. On page 7, it's said "The length of future context tokens (m) is 50 and 10 for FINE-Mask and FINE-Wait, respectively." How to select the value of m, and why is it different for both?
3. Do authors think when training is conducted in a streaming manner (e.g. similar as FINE-Mask and FINE-Wait), accuracy could be further improved with streaming inference? If so, is there a way to train speech translation models using streaming inputs, and generalize it for different streaming inference configurations?
4. In the last section of conclusion, it's said "In the future, we will experiment with other methods to improve the accuracy of predicting future information." This is not informative. It would help to list examples for methods/directions authors think worth exploring, if applicable.

---

> ### Author Response · Authors · 2022-11-18
> **Thanks for the insightful comments. Here is the response to Reviewer 2wmu.**
>
> Thanks for the insightful comments. We would like to address the novelty of this submission. Most prior works on streaming ST try to tackle the streaming inference via training in a streaming (or online) fashion as well. Particularly, if some methods require the fixed k or N (in wait-k or stride-N) during training, the inference should also use the same k or N. To meet the requirement of different inference latency, different ST models are trained in different k and N. However, we're working on another research direction: how to adapt a single pre-trained offline ST model to streaming inference without further streaming training, and how to close the gap between offline pre-training and online inference. The topic itself is novel though our solution is simple. Particularly, our FINE-Mask strategy is a simple yet effective solution.
>
> > Q1: Do authors have comments on model size and computation cost, when comparing the proposed methods vs. baselines?
>
> A1: We didn't comment based on two reasons. First, MoSST as our base model, has the exactly the same model size and computation cost. Other baselines (SimulSpeech, RealTrans, MU-ST) didn't reveal the model size in the paper or open-source their checkpoint. Second, our streaming ST is only a single offline ST model. However, other baselines (SimulSpeech, RealTranS) are trained in a streaming manner. Evaluating them in different latency actually requires multiple models. So it is difficult for a fair comparison.
>
> > Q2: How to select the value of m, and why is it different for both?
>
> A2: We have already mentioned in ablation study Sec 5.3.1.
>
> > Q3: Do authors think when training is conducted in a streaming manner (e.g. similar as FINE-Mask and FINE-Wait), accuracy could be further improved with streaming inference?
>
> A3: The main research topic in this work is how to reuse a pre-trained offline ST model for streaming inference without further streaming related training, and how to close the gap between offline pre-training and online inference. But we agree if training in a streaming manner, the performance on streaming inference should be improved, and this falls into the traditional research direction of training streaming ST. In the next question, we will talk more about this direction.
>
> > Q4: "In the future, we will experiment with other methods to improve the accuracy of predicting future information." is not informative. list examples for methods/directions authors think worth exploring.
>
> A4: Our research direction is to adapt pre-trained offline ST to online inference. As mentioned, the main gap between them is the speech features in streaming input at the last several positions are inferior. So we use FINE-Mask to improve the feature quality. But according to our analysis in Appendix B.8, if we replace partial speech features to the truncated full speech features, the streaming inference performance can be greatly improved.  Therefore, when we said "improve the accuracy of predicting future information", it means that whether we can use mask tokens to predict better future speech features which are similar to the full speech feature.

---

### Official Review · Reviewer_xgfh · 2022-10-27

**Confidence:** 4
**Correctness:** 3
**Technical Novelty And Significance:** 2
**Empirical Novelty And Significance:** 3
**Recommendation:** 5

**Clarity, Quality, Novelty And Reproducibility:**

Clarity
- The paper is written in general.
- Since I have more ASR backgrounds (and many other people interested in this work would be the same backgrounds), I think it is better to mention that this paper deals with an attention-based method and briefly describes other streaming speech processing based on RNN-T and CTC in the beginning.
- The end of the introduction discusses performance improvement, but since it does not clearly mention the baseline, I could understand the significance of this improvement. It is better to briefly explain the baselines for the readers to understand the importance of this method.
- MoSST seems to be an important base model of paper, but it does not explain it clearly. Also, it is better to explain why the paper adopts MoSST as a base model.
- Section 3, 3rd paragraph suddenly uses "a factor 320," which is confusing. I expect the paper to assume 16kHz sampling and 320 samples (corresponds to 20 ms frameshift) in W2V2. It is better to clarify it.

Quality
- This work has a solid analysis to motivate the proposed method. This study's quality is sufficient with the proposed method's apparent effectiveness.

Novelty
- I'm not entirely convinced of this method in terms of the significant novelty for the machine learning community since the method attaches special tokens or waiting schemes. They are very logical based on their analysis, but the techniques are not significantly novel.
- If there are more technical novelty, solid theoretical background, etc., for the FINE-Mask/FINE-wait, I would recommend the authors spend more space explaining them in more detail.

Reproducibility
- The database and evaluation metrics are publically available. The reproducibility is high.

**Strength And Weaknesses:**

Strength
- Streaming speech translation becomes an essential technique for conversational AI.
- Clear analysis of the streaming issue in speech translation by focusing on the end position. I like their formulation of the w2v2 output representation for streaming and offline based on two-time indexes $a _{t, t}$ (but it requires a bit more explanation). This makes it easy to understand the feature representation in the end position.
- FINE-mask/wait are simple but powerful
- The paper is well written overall.

Weaknesses
- Although this would be a strength, I think the method is a bit too simple and does not have as much algorithmic/theoretical novelty as a machine learning conference.
- I want to have more relationships with streaming speech recognition techniques. I know that some methods (e.g., RNN-T) cannot be straightforwardly used for speech translation due to the reordering issues, but there are several attention-based streaming ASR methods (e.g., monotonic attention-based approaches like mocha https://arxiv.org/abs/1712.05382).
- It is better to have more challenging language pairs with more serious reordering issues, e.g., En-Ja is recently supported in Must-C.

**Summary Of The Paper:**

This paper proposes a new algorithm for streaming speech translation by improving the wait k policy-based approach with a mask trick. The paper first starts with the streaming speech translation issue caused by the end-position speech feature representations with various analyses. This analysis naturally motivates their proposed method of using mask tokens or extra speech tokens to avoid dealing with unstable end-position speech feature representations. The experiment is based on a public speech translation corpus (MustC). It shows the effectiveness of the proposed method based on the latency and translation performance trade-off with several ablation studies.

**Summary Of The Review:**

Although the paper is clearly written with its rationale for solving the end-position issue by simple speech token augmentations, I feel that the technical novelty of this paper is not significant due to the above reasons.

Several other suggestions and questions:
- How about applying this technique to steaming ASR? This will prove the generalization of this work and would attract more machine learning researchers
- I thought that $t'=t$ in Fig.2 (c) has the highest cosine similarity compared with the other two examples since this streaming condition is the same as the offline case (I think I misunderstood something). Can you explain it?

---

> ### Author Response · Authors · 2022-11-18
> **Thanks for your insightful comments. Here is the response to the weaknesses for Reviewer xgfh.**
>
> > Q1: Although this would be a strength, I think the method is a bit too simple and does not have as much algorithmic/theoretical novelty as a machine learning conference.
>
> A1: Most prior works on streaming ST try to tackle the streaming inference via training in a streaming (or online) fashion as well. Particularly, if some methods require the fixed k or N (in wait-k or stride-N) during training, the inference should also use the same k or N. To meet the requirement of different inference latency, different ST models are trained in different k and N. However, we're working on another research direction: how to adapt a single pre-trained offline ST model to streaming inference without further streaming training, and how to close the gap between offline pre-training and online inference. The topic itself is novel though our solution is simple. Particularly, our FINE-Mask strategy is a simple yet effective solution.
>
> > Q2: I want to have more relationships with streaming speech recognition techniques. I know that some methods (e.g., RNN-T) cannot be straightforwardly used for speech translation due to the reordering issues, but there are several attention-based streaming ASR methods (e.g., monotonic attention-based approaches like mocha https://arxiv.org/abs/1712.05382).
>
> A2: Most streaming recognition techniques still fall into the category to train the model in a streaming manner, e.g., MoChA. Similalry, CAAT (https://arxiv.org/pdf/2107.00279.pdf) follows the RNN-T method to build a streaming speech translation model. \
> However, what we are researching is completely different from these streaming recognition techniques. We want to reuse the pre-trained offline ST model for streaming inference without further streaming-related training. In general, this is a difficult problem for two reasons. **First**, the streaming inference takes the partial speech sequence as input, while the offline model always consumes the full speech sequence. **Second**, the reordering issue as you mentioned prevents partial speech sequence from seeing future information. Our proposed method attempts to relieve such mismatchs via the FINE-Mask strategy.
> In summary, our method is different from traditional streaming techniques. To make readers better understand, we have added a relevant description in Appendix C of revised version. Thanks for the suggestions.
>
> > Q3: It is better to have more challenging language pairs with more serious reordering issues, e.g., En-Ja is recently supported in Must-C.
>
> A3: We have a similar experiment on En-De (See Table 1 and Fig 10a in Appendix). We divided the En-De testset into 3 subsets accordding to their monotonic level (i.e., reordering level). Indeed, for more monotonic data, our model performs better and the BLEU is close to offline model. For data with more reordering, our model performance drops and the gap to offline model is large.

---

> > ### Author Response · Authors · 2022-11-18
> > **Thanks for your valuable comments. Here is the response to the clarity for Reviewer xgfh.**
> >
> > Thansk for the suggestions in clarity. We have modified the submission accordingly.
> >
> > > Q4: Why MoSST as a base model.
> >
> > A4: There are two reasons. **First**, as Sec 4.1 mentioned, the acoustic encoder is pre-trained wav2vec2.0. The mask speech embedding is well-trained and is helpful for our FINE-Mask straregy. **Second**, as the Sec 5.1 mentioned, MoSST has a continuous integrate-and-fire (CIF) module that can be trained to roughly segment the speech sequence into words level with very weak signal (the total number of words). After training, the number of resulted segments is roughly equal to the number of words. So it can be used as a pseudo word detector for streaming inference.
> >
> > > Q5: How about applying this technique to steaming ASR?
> >
> > A5: In general, it is beyond the scope of this paper as we're not focusing on pure streaming techniques. We believe that our technique is promising for turning an offline ASR into streaming ASR. But we think at least two modifications are required when training the offline ASR, wav2vec2.0 pretraining and CIF module.
> >
> > > Q6: $t^{\prime}=t$ has the highest cosine similarity compared with the other two examples.
> >
> > A6: For clarity, let's take an example to describe the calculation process in Figure 2(c).
> > For a full speech $\mathbf{x}=(x_1,x_2,x_3)$ and its representation $\mathbf{a}=(a_1,a_2,a_3)$, there are 3 streaming inputs and 3 corresponding representations:
> >
> > - $\hat{\mathbf{x}}\_{t=1}=(x_1)$ and $\hat{\mathbf{a}}\_{t=1}=(a\_{t=1,t^{\prime}=1})$
> > - $\hat{\mathbf{x}}\_{t=2}=({x_1,x_2})$ and $\hat{\mathbf{a}}\_{t=2}=(a\_{t=2,t^{\prime}=1},a\_{t=2,t^{\prime}=2})$
> > - $\hat{\mathbf{x}}\_{t=3}=(x_1,x_2,x_3)$ and $\hat{\mathbf{a}}\_{t=3}=(a\_{t=3,t^{\prime}=1},a\_{t=3,t^{\prime}=2},a\_{t=3,t^{\prime}=3})$
> >
> > the cosine similarity at every $t^{\prime}=t$ position is calculated as: $\frac{1}{3}(\operatorname{cos}(a_{t=1,t^{\prime}=1},a_1)+\operatorname{cos}(a_{t=2,t^{\prime}=2},a_2)+\operatorname{cos}(a_{t=3,t^{\prime}=3},a_3))$\
> > the cosine similarity at every $t^{\prime}=t-1$ position is calculated as: $\frac{1}{2}(\operatorname{cos}(a_{t=2,t^{\prime}=1},a_1)+\operatorname{cos}(a_{t=3,t^{\prime}=2},a_2))$
> >
> > Full speech feature $a_t$ is obtained by always attending to full speech tokens. \
> > When calculating partial speech feature $a_{t,t'=t}$, for current position $t'$, $a_{t,t'=t}$ is obtained by attending to speech tokens ($\leq t=t'$). \
> > When calculating partial speech feature $a_{t,t'=t-1}$, for current position $t'$, $a_{t,t'=t-1}$ is obtained by attending to speech tokens ($\leq t=t'+1$). In other words, $a_{t,t'=t-1}$ can see one more future speech token, so $\operatorname{cos}(a_{t,t'=t}, a_t)$ is usually higher than $\operatorname{cos}(a_{t,t'=t}, a_t)$.

---

### Official Review · Reviewer_uAxq · 2022-10-31

**Confidence:** 4
**Clarity, Quality, Novelty And Reproducibility:** 1) **Clarity and Quality**
**Correctness:** 4
**Technical Novelty And Significance:** 3
**Empirical Novelty And Significance:** 3
**Recommendation:** 6

**Strength And Weaknesses:**

**Strengths**
1) Strong motivation with supported analysis for the motivation and a simple solution towards this problem.
2) The paper is well written and clear to understand.

**Weaknesses**
1) It seems like the approach is limited to wav2vec type models from its motivation. But in my opinion this approach should work for any kind of models, specially the FINE-Wait. A study or discussion on this would have made the paper stronger.
2) The paper is also missing details/analysis on how does number of fine-tuning steps affects the FINE-Mask strategy. If the model is reliant on the MASK tokens that are used during self-supervised pre-training I would be curious to learn if amount of fine-tuning causes a change in the performance.
3) Similarly, the approach could have been stronger if the authors also studied how these model training could be modified to match the inference conditions. For example, maybe providing future biased masking during fine-tuning would allow the models to match the inference conditions.

**Summary Of The Paper:**

This paper addresses the differences between the training and inference conditions of streaming models. They first present a thorough analysis of the problem by finding correlation between representations during offline and streaming modes and plot that against the model performance in terms of BLEU score. They show that the deterioration happens from the last representations of the input.

In order to fix this problem they propose a simple strategy during inference to match the conditions that wav2vec 2.0 based models are trained towards. They propose two strategies FINE-Wait and FINE-Mask and also discuss a combination of it called FINE-Hybrid. These techniques essentially create a pseudo longer context using mask tokens (matching the wav2vec 2.0 training criterion) or using actual speech tokens for better contextualization and then dropping it during inference.

Overall, the technical contribution is low but the idea is neat and simple with a good motivational analysis. The drawback is that it's quite constrained towards models trained on wav2vec 2.0 criterion.

**Summary Of The Review:**

I think the paper is well motivated, simple yet strong, if the authors are able to comment on the weaknesses I pointed out, this paper can be quite insightful.

---

> ### Author Response · Authors · 2022-11-18
> **Thanks for the insightful comments. Here is the response to Reviewer uAxq.**
>
>
> > Q1: Do FINE-Mask and FINE-Wait work for non-wav2vec type models?
>
> A1: FINE-Mask should not work for non-wav2vec type models, because mask token embedding are pre-trained in wav2vec2.0. \
> For FINE-Wait, it is possible to work. We have a quick experiment to test an offline NMT model. With the same wait-k policy, FINE-Wait inference by peeking 1 or 2 tokens can achieve a signifcant BLEU improvement over regular inference. However, when considering the AL, the BLEU vs AL curve of two inference methods are highly coincident. It should be a great future work to explore this direction on text NMT.
>
> > Q2: How does number of fine-tuning steps affects the FINE-Mask strategy?
>
> A2: We didn't set the number of fine-tune steps, but wait the model to converge on the validaton set.
> In this response, we conduct another experiment to verify this concern. We evaluate the FINE-Mask inference and regular streaming inference at different checkpoints with epoch=23,25,27,29,31 after training converges. We can observe that FINE-Mask consistently brings improvement over the baseline. The detailed experiments can refer to the Fig. 15 in Appendix of the revised vision.\
> In the original submission, a relevant experiment is conducted in the Appendix B.4 and the correspding Figure 11. We've compared the quality of predicted speech features in FINE-Mask setting between finetuned wav2vec2 and the pre-trained one on both MuST-C and Librispeech testsets. Surprisingly, we found the fine-tuned version is always better than the pre-trained one.
>
> > Q3: The approach could have been stronger if the model training could be modified to match the inference conditions.
>
> A3: The main contribution is to apply online ST inference with the pre-trained offline model without any modification. So the offline ST model can do both offline and online inference. But we agree that if the modification of offline training to match online inference can keep the offline performance unchanged, it will be a great research direction for future work.\
> Meanwhile, we have discussed the performance upper bound of the FINE-Mask in the Appendix B.8.
> As mentioned, the main gap between offline training and streaming inference is the speech features in streaming input at the last several positions which are inferior. So we use FINE-Mask to improve the feature quality. But according to the analysis in B.8, if we replace partial speech features to truncated full speech feature, the streaming inference performance can be greatly improved.  Therefore, when we said "improve the accuracy of predicting future information" in the conclusion as the future work, it means that whether we can use mask tokens to predict better future speech features which are similar to the full speech feature.
>
> > Q4: Novelty : Low, yet a simple and effective approach.
>
> A4: We would like to address the novelty of this submission. Most prior works on streaming ST try to tackle the streaming inference via training in a streaming (or online) fashion as well. Particularly, if some methods require the fixed k or N (in wait-k or stride-N) during training, the inference should also use the same k or N. To meet the requirement of different inference latency, different ST models are trained in different k and N. However, we're working on another research direction: how to adapt a single pre-trained offline ST model to streaming inference without further streaming training, and how to close the gap between offline pre-training and online inference. The topic itself is novel though our solution is simple. Particularly, our FINE-Mask strategy is a simple yet effective solution.
>
> > Q5: Reproducibility : Low, no details on the model was trained.
>
> A5: We add more detailed training details in Appendix D of our revised version. In addition, we also provide code in the supplementary material, which contains detailed training and inference settings.

---

### Decision · Program_Chairs · 2023-01-20

**Decision:**

Reject

**Justification For Why Not Higher Score:**

As noted in the summary, the proposed future-aware method is a widely used method in speech processing, for both automatic speech recognition and speech translation. Although the FINE-MASK method has its novelty, it is not generalized to other pretraining models. Furthermore, the proposed method is too speech-specific, not generalized to other areas.

**Justification For Why Not Lower Score:**

N/A

**Metareview: Summary, Strengths And Weaknesses:**

This paper proposes a new algorithm for streaming speech translation model by utilizing the information from future speech frames. The algorithm starts from an offline pre-trained wav2vec 2.0 model, and uses FINE-Mask and FINE-Wait to improve the translation quality. The major contribution of this paper is to show the importance of future speech frames with wav2vec 2.0 as a baseline offline model.  All reviewers believe this is a very well written paper. The analysis part is also very informative.

However, the novelty is the biggest concern from reviewers. Utilizing future frames is not new in streaming speech tasks. There are lots of works in the speech recognition area using future frames with streaming Transformers. Here are some examples
[1] Zhang, Qian, et al. "Transformer transducer: A streamable speech recognition model with transformer encoders and rnn-t loss." ICASSP 2020-2020 IEEE International Conference on Acoustics, Speech and Signal Processing (ICASSP). IEEE, 2020.
[2] Shi, Yangyang, et al. "Emformer: Efficient memory transformer based acoustic model for low latency streaming speech recognition." ICASSP 2021-2021 IEEE International Conference on Acoustics, Speech and Signal Processing (ICASSP). IEEE, 2021.
[3] Kim, Kwangyoun, et al. "Multi-mode Transformer Transducer with Stochastic Future Context." arXiv preprint arXiv:2106.09760 (2021).
[4] Mahadeokar, Jay, et al. "Flexi-Transducer: Optimizing Latency, Accuracy and Compute for Multi-Domain On-Device Scenarios." in Proc. Interspeech, 2021.

The authors argued that the traditional methods can only have fixed N or fixed K in stride-N or stride-K to match training and testing. However, methods in [3][4] can have arbitrary N during inference for the same model. There are also streaming speech translation works that don’t have fixed policy by leveraging RNN-T, such as CAAT or
[5] Xue, Jian, et al. "Large-Scale Streaming End-to-End Speech Translation with Neural Transducers." in Proc. Interspeech, 2022.
In [5], future speech frame is also used to improve speech translation quality.
Look-ahead is also used in some related works mentioned by the authors:
[6] Chen, Junkun, et al. "Direct Simultaneous Speech-to-Text Translation Assisted by Synchronized Streaming ASR." Findings of the Association for Computational Linguistics: ACL-IJCNLP 2021. 2021.
[7] Zeng, Xingshan, et al. "RealTranS: End-to-End Simultaneous Speech Translation with Convolutional Weighted-Shrinking Transformer." Findings of the Association for Computational Linguistics: ACL-IJCNLP 2021. 2021.
As a conclusion, utilizing future speech frames is a natural choice for streaming speech recognition and translation tasks. Therefore, the novelty is really limited.

The FINE-Mask is a very interesting idea that applies to wav2vec 2.0, which uses masked tokens. However, this may not be applicable to other pretraining methods, like standard ASR pretraining. Ideally, the proposed method should not be restricted to only one pretraining method in order to have larger impact.

For a paper published in ICLR, ideally the paper should have larger audience. This paper focuses too specifically for speech. If the paper is submitted to ICASSP or Interspeech, it may attract more attention, especially if the authors can strengthen the novelty by generalizing the proposed method not only on the pre-trained wav2vec 2.0 but also to other popular pre-trained speech models without using masked tokens.


**Summary Of Ac-Reviewer Meeting:**

All the reviewers agreed the paper is clearly written. All reviewers agreed that the paper has the novelty of leveraging pre-trained offline model for streaming speech translation. The FINE-MASK method is novel. The proposed method is simple but effective.

The novelty is the biggest concern. Note that the reviewer of the highest score is not familiar with the speech domain, therefore, his judgement is based on the paper writing. In contrast, reviewers who work in the speech domain pointed out the paper novelty is limited because leveraging future frames to improve model quality is a widely used method is speech. The reviewers are not satisfied with the reply of the authors on the novelty.
A reviewer pointed out the ICLR paper should benefit audience from multiple domains instead of only in the speech domain. The audience in the speech conferences (ICASSP/Interspeech) may appreciate this work more.